# Using Neural Networks to Obtain Indirect Information about the State Variables in an Alcoholic Fermentation Process

Anca Sipos [1,*], Adrian Florea [2], Maria Arsin [2] and Ugo Fiore [3]

1. Lucian Blaga University of Sibiu, Faculty of Agricultural Sciences, Food Industry and Environmental Protection, 7–9 Dr. Ion Ratiu Street, 550012 Sibiu, Romania
2. Department of Computer Science, Lucian Blaga University of Sibiu, Faculty of Engineering, 4 Emil Cioran Street, 550025 Sibiu, Romania; adrian.florea@ulbsibiu.ro (A.F.); maria.arsin15@gmail.com (M.A.)
3. Department of Management and Quantitative Studies, Parthenope University of Napoli, 80132 Napoli, Italy; ugo.fiore@uniparthenope.it
* Correspondence: anca.sipos@ulbsibiu.ro

**Abstract:** This work provides a manual design space exploration regarding the structure, type, and inputs of a multilayer neural network (NN) to obtain indirect information about the state variables in the alcoholic fermentation process. The main benefit of our application is to help experts reduce the time needed for making the relevant measurements and to increase the lifecycles of sensors in bioreactors. The novelty of this research is the flexibility of the developed application, the use of a great number of variables, and the comparative presentation of the results obtained with different NNs (feedback vs. feed-forward) and different learning algorithms (Back-Propagation vs. Levenberg–Marquardt). The simulation results show that the feedback neural network outperformed the feed-forward neural network. The NN configuration is relatively flexible (with hidden layers and a number of nodes on each of them), but the number of input and output nodes depends on the fermentation process parameters. After laborious simulations, we determined that using pH and $CO_2$ as inputs reduces the prediction errors of the NN. Thus, besides the most commonly used process parameters like fermentation temperature, time, the initial concentration of the substrate, the substrate concentration, and the biomass concentration, by adding pH and $CO_2$, we obtained the optimum number of input nodes for the network. The optimal configuration in our case was obtained after 1500 iterations using a NN with one hidden layer and 12 neurons on it, seven neurons on the input layer, and one neuron as the output. If properly trained and validated, this model can be used in future research to accurately predict steady-state and dynamic alcoholic fermentation process behaviour and thereby improve process control performance.

**Keywords:** neural network; fermentation process; prediction application

## 1. Introduction

We live in an age of advanced technology from both a hardware and a software viewpoint, the so called "centaur era" in which man and machine work together alongside reality-augmenting computers to assist humans [1]. The ubiquity of computing shows that technology is not expressed only as laptops, computers, or tablets but is also found in almost everything that surrounds us, such as in communications, banking, and transport; the food industry is no exception.

If the physical and chemical conditions (temperature, pH, aeration, etc.) are favourable, a biotechnological process can induce the growth of a microorganism population, called a biomass, in a vessel through the consumption of some nutrients (carbon, nitrogen, oxygen, vitamins, etc.) that represent the substrate [2]. In the vessel (bioreactor), many biochemical and biological reactions take place simultaneously. Usually, each elementary reaction is catalysed by a protein (enzyme) and can form a specific product or metabolite. The aim of such a process can be the production of bacteria, yeasts, etc.; the development of particular

components (amino acids, medicines, marsh gas, etc.); or biological decontamination (the biological consumption of a polluted substrate by the biomass).

The alcoholic fermentation process is a biotechnological process and is undoubtedly one of the most important steps in winemaking [3,4]. The alcoholic fermentation in the winemaking industry is a complex process that must account for particular characteristics, including the following: batch fermentation on natural complex media, anaerobic conditions due to $CO_2$ production, the composition of the raw material, the low media pH, levels of sulphur dioxide, inoculation with selected yeasts, and interactions with other microorganisms.

Controlling the alcoholic fermentation process is a delicate task in winemaking for several reasons: the process's complexity, nonlinearity, and non-stationarity, which make modeling and parameter estimation particularly difficult, and the scarcity of on-line measurements of the component concentrations (essential substrates, biomass, and products of interest) [5,6]. One of the core issues in industrial winemaking involves developing soft sensors with outstanding performance and robustness to replace the hardware/physical sensors in bioreactors. This would mitigate the disadvantages of real-time measurements, nonlinearity constraints, and other complex mechanisms in the fermentation process [7].

An alternative to overcome the difficulties mentioned above is to use neural networks (NNs), one of the fastest growing areas of artificial intelligence. With their massive learning capabilities, NNs are able to approximate any continuous functions [8,9] and can be applied to nonlinear process modeling [10,11]. If properly trained and validated, these models can be used to accurately predict steady-state and dynamic process behaviour and thus improve process control performance [12,13].

A NN model can offer information regarding the values of the state variables (as inputs: biomass and temperature from the bioreactor; as outputs: alcohol and the substrate) useful for a control system in the fermentation process. This is is due to the ability of a NN to "learn" the shape of a relationship between variables from the data observed in the training regime and generalize that relationship to the data zone requested in the test regime. Such information is important especially in the exponential phase of biomass production [14].

Most of the scientific literature indicates that because of the complexity of biotechnological systems regarding alcoholic fermentation, traditional optimization methods utilizing mathematical models and statistically designed experiments are less effective, especially on a production scale. Furthermore, Machine Learning (ML) offers an effective tool to predict biotechnological system behavior and empower the Learn phase in the Design-Build-Test-Learn cycle used in biological engineering [15]. NNs provide a range of powerful techniques for solving problems in sensor data analysis, fault detection, process identification, and control and have been used in a diverse range of chemical engineering applications [16]. Moreover, according to [17], even though both NN and RSM (Response Surface Methodology) can efficiently model the effects of the interactions between the input and the output parameters, the NN model is more robust for predictions in non-linear systems. RSM disregards the "less important" variables based on a limited understanding of their possible interactive effects on the bioprocess output. Since the number of inputs in our experiment is not very large, we did not consider it appropriate to use RSM, which prunes the design space, negatively affecting the results.

The main aim of this paper was to develop an application that can predict the characteristic variable evolution of a system in the food industry to obtain indirect information about the process regarding the state variables usable in an advanced control system for the alcoholic fermentation batch process of white wine as a knowledge-based system. To achieve this goal, we developed this study in the MATLAB environment and trained a NN. We used the experimental data for an alcoholic fermentation process for white winemaking and then, based on this NN, predicted the desirable variables for this process.

In this context, we created and trained different types of neural networks: feedforward and feedback. After implementation, the application was tested on different

configurations of NNs to find the optimal solution from the perspectives of prediction accuracy and simulation time.

The software application predicts the values of the variables that characterise the alcoholic fermentation process of white wine to help food industry specialists more easily control this process in winemaking and reduce production costs. The main benefit of our application is that it can help experts, thereby reducing the need for many time-consuming measurements and increasing the lifecycles of sensors in bioreactors. The novelty of this research lies in the flexibility of its applications, the use of a great number of parameters, and the comparative presentation of the results obtained with different NNs (feedback vs. feed-forward) and different learning algorithms (Backpropagation vs. Levenberg–Marquardt).

The application development stages are as follows:

- Characterizing the alcoholic fermentation process and its phases and mapping them to the software
- Recording the values of the state variables in alcoholic fermentation in a database that will be used in the prediction application
- Setting-up, training, and tuning a NN with the data obtained from the fermentation processes
- Using the trained NN to predict, in different situations, the values of some variables that characterise the alcoholic fermentation process.

The rest of the paper is organized into four sections. Section 2 is split into two parts: The first provides a short background of NNs and their learning algorithms, while the second briefly reviews state-of-the-art papers related to this study. Section 3 describes the proposed approach—the materials and methods—for modelling the alcoholic fermentation process in making white wines, implementing the NNs and measurement data. Section 4 analyses the experimental results obtained, providing some interpretations and possible guidelines. Finally, Section 5 highlights the paper's conclusions and suggests future research directions.

## 2. Related Work

### 2.1. Short Background of NNs

From a purpose point of view, NNs can be viewed as part of the larger domain of pattern recognition and Artificial Intelligence [18]. From the point of view of the method applied, NNs fall within the parallel distributed processing domain. Generally, artificial NNs try to simulate the neurophysiological structure of the human brain. The cortex is composed of a large number of interconnected biological cells called neurons. Each neuron receives signals from the neurons connected to it through the dendrites and conveys a signal using the axon. A simple mathematical model for this process considers the output of a unit as a function (the activation function, usually nonlinear) of an affine transformation of the outputs of the connected units. The coefficients of the transformation (the intercepts are called biases; the other coefficients are called weights) determine the response and can be adapted depending on how the connection is activated during training. A NN is an information processing system composed of a multitude of units (neurons) that are strongly interconnected. Even though biological neurons are slower than logical gates implemented in silica, neurons can accomplish tasks that are beyond the reach of the best computers currently available. The brain compensates for the relatively slow operations of its individual components by means of a large number of interconnections that are flexible and malleable, allowing them to adapt to the environment, to handle vague or imprecise information, and to generalise based on known situations and examples of unknown situations in a robust, error-tolerant way [19].

#### 2.1.1. NN Types

There are several types of artificial NNs classified according to different factors:

1.　The topological structure of the neurons:

- single-layer networks
- multilayer networks

2. The direction in which the signals flow:
   - feed-forward networks
   - feedback networks

In NNs, units are arranged in layers such that units in the same layer do not interact, while the outputs of units in a layer are used as inputs into the units in the adjacent layer. In **Single-layer networks**, there is only one layer; here, the inputs of the neurons are the inputs of the entire network, and the outputs of the neurons are the outputs of the network. **Multilayer networks** have more layers divided into three categories: the input layer, the middle (hidden) layers, and the output layer.

**Feed-forward networks** (see Figure 1) are NNs in which the signal can only propagate unidirectionally. In these networks, the output vector can be determined by direct calculations from the input vector.

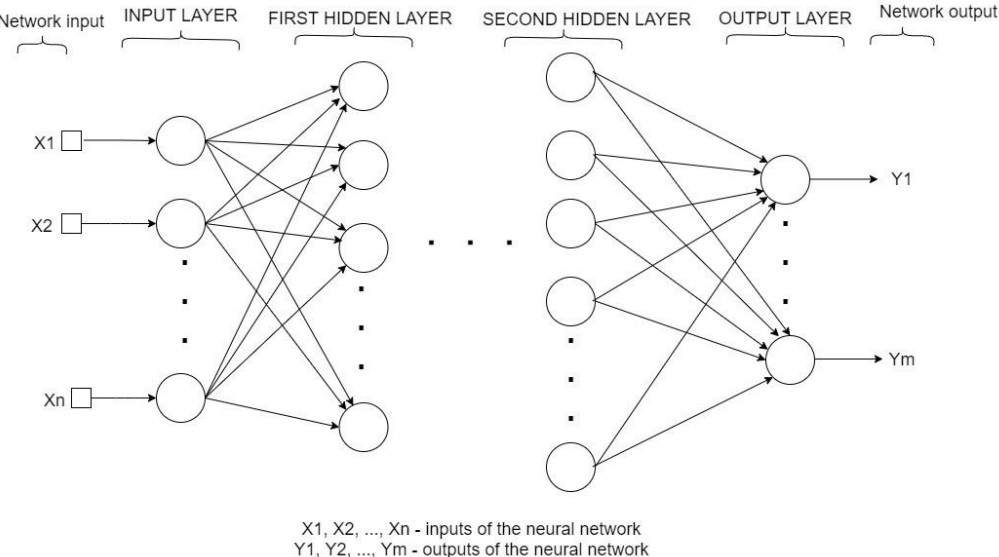

Figure 1. The architecture of a generic multilayer neural network with two hidden layers.

**Feedback neural networks** (bidirectional or recurrent) are artificial NNs in which the connections from the neurons are cyclical, allowing the signal to be conveyed in both directions. This type of network uses a previous output as an additional input to calculate the next output. In other words, the units of such networks have an internal state.

Recurrent networks (see Figure 2) are generally operated to classify data that form sequences, such as in text and images processing. Each element of the word/characters/images sequence can depend on the framework created by the previous elements of the sequence, but these networks are used also to predict some values from temporal data series (data processed at successive time points) [13,19].

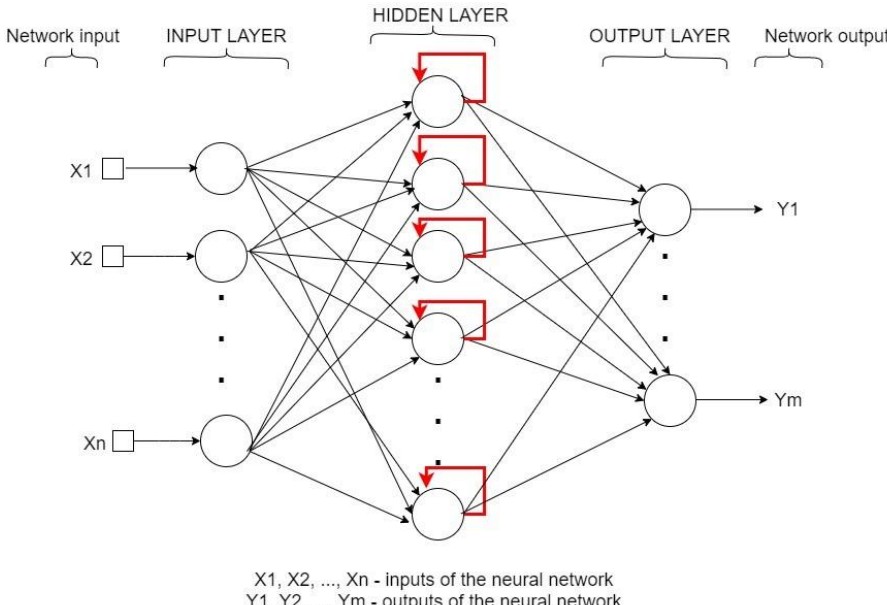

X1, X2, ..., Xn - inputs of the neural network
Y1, Y2, ..., Ym - outputs of the neural network

**Figure 2.** The architecture of a generic feed-back (recurrent) neural network with a single hidden layer.

### 2.1.2. The NNs Learning Mechanism

The learning capacity of NNs is one of their fundamental features. During training, weights are adjusted based on the input and the output so that future network behaviour will be consistent with previous experience. The training considered here will be of the supervised type, meaning that the training process is based on some practice examples in the form of input–output pairs. The network generates an output that is then compared with the desired output from the practice set. If the generated output of the network does not coincide with the desired output, it will be necessary to modify the weights and biases from all layers of the network, starting with the output layer and working towards the input layer. By contrast, in unsupervised learning, no labelled data are presumed to be available [13].

To minimize the input error between the value generated by the network and the desired value (the goal value) for a given collection of training examples, the direction in which weights should be modified needs to be computed. The Backpropagation (BP) algorithm uses the chain rule to compute the gradient of the error for each unit with respect to its weights [20]. Then, an algorithm is needed to optimise the weights. Note that performance is important because the changes induced by each training example need to be smoothened and combined over the entire training set, which involves multiple runs. The Levenberg–Marquardt (LM) algorithm is an iterative optimizational technique that uses aspects of the gradient descent and Gauss-Newton method and is fast in practice, as will be demonstrated in our experiments. The balance between these two methods is governed by a damping term: When the damping term is large, the algorithm behaves like the steepest descent method—slow but able to cope with highly nonlinear regions; when the damping term is small, the LM step approximates the Gauss-Newton step for faster convergence. The LM algorithm adaptively adjusts the damping term, reducing it when the step effectively reduces the error and increasing it when it fails to do so [21,22].

### 2.2. Modeling the Alcoholic Fermentation Process of the White Wines with NN

Fermentative bioprocesses (like alcoholic fermentation) are basic processes in food engineering [23]. However, because they are biological processes with complex and strongly variable behaviour, the phenomena that govern them are poorly known, their significant state variables are difficult to define, and their dynamic behaviour is generally strongly

nonlinear. Consequently, the experiments carried out within relevant studies are long and difficult to reproduce and the information concerning the state variables of the process are difficult to obtain—usually by sampling and laboratory analyses due to a lack of adequate sensors. Thus, in most cases, indirect measurements and determinations are needed.

A good solution to obtain indirect information from a process consists of building a neuronal model, which—based on the experimental data available—can offer information concerning the values of state variables, which can then be used in an automation control system [2].

To realise a software application that can predict the evolution of the state variables over time, the type of NN most suitable for this purpose must first be determined. In the literature, most articles suggest the use of feed-forward NNs with the Levenberg–Marquardt learning algorithm [7,24].

As specified in Section 2.1, a NN can be feed-forward or recurrent. The latter NNs are specifically used for data that show a strong dependence on each other—particularly text sequences, pixels of an image, or time series data. Based on this observation, we also chose to implement in our software application the possibility to predict the desired values with a recurrent network because the experimental data that will be used to train the network are a time series. The alcoholic fermentation process is a process that modifies the values of the variables that characterise the grapes' juice as a function of time and the current values of these variables, so using a recurrent network is justified.

## 3. Materials and Methods

The organization of the experiments began with defining the process variables (according to the activity diagram from the scheme presented in Figure 3). Based on a literature review, we decided that the most appropriate tool to obtain indirect information about the state variables in an alcoholic fermentation process is a NN. The next step was to follow a manual design space exploration to determine the structure, type, and inputs of the multilayer neural networks. The trial and error process was time-consuming and aimed to reduce the prediction errors. We first applied original data and then used normalized data. As inputs for the process, pH and $CO_2$ were found to be the best options.

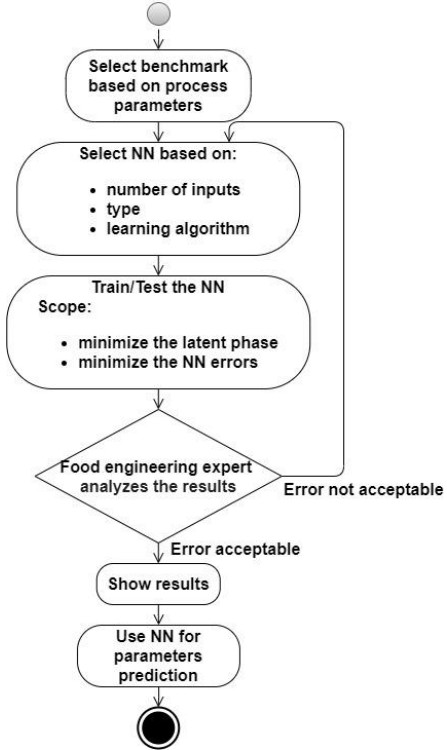

**Figure 3.** The scheme of the experiments' organization.

Twenty datasets of experimental processes were used for the analysis, which was done in the bioreactor in the research laboratory. The investigation of the experimental data was conducted under four distinct control situations based on the substrate: mash malted, mash malt enriched with vitamin B1 (thiamine), white grape must, and white grape must enriched with vitamin B1 (thiamine). For the experiments, *Saccharomyces oviformis* and *Saccharomyces ellipsoideus* wine yeasts were seeded on a culture medium [25]. The bioreactor was equipped with pH, temperature, level, and stir speed controls, as well as dissolved $O_2$, released $CO_2$, and $O_2$ sensors and analysers. The cell concentration was calculated based on three different parameters: optical density, dry substance, and total cell number. The ethanol concentration was determined using HPLC-MS equipment, and the substrate concentration was determined by spectrophotometric techniques. The operating conditions were as follows: working volume—8 L; temperatures of 20, 22, 24, 26, and 28 °C; stirring speed—150 rpm; pH—3.8; and influent glucose concentrations—180 g/L and 210 g/L. The necessary oxygen was dissolved in must without aeration.

A controlling solution for the alcoholic fermentation process was developed using a NN to obtain indirect information about the process. Figure 4 describes the general framework using both the bioreactor and the software application to predict the process control parameters. The NN used to predict the desired variables of the process was trained with experimental data obtained from the fermentation process taking place in the bioreactor equipped with transductors or by acquiring samples during fermentation and analyzing them in the laboratory. The dataset contains the values of the following variables that characterise the alcoholic fermentation process:

- $T$ fermentation temperature [°C]
- $t$ time [h]
- $S_0$ the initial concentration of the substrate [g/L]
- $S$ the substrate concentration [g/L]
- $X$ the biomass concentration [g/L]
- $P$ the alcohol concentration (product) [g/L]
- the mixing speed [rpm];
- the optical density of the mass fermentation [AU]
- the $pO_2$ [%]
- the pH
- the released $O_2$ concentration [volume %]
- $C_{CO_2}$ the released $CO_2$ concentration [volume %].

Based on all the training experimental data, we used four input–output configurations for the NN, as presented in Table 1. These configurations contained the values of the variables necessary to control the process and also some supplementary values to determine a version where the prediction error is minimal.

**Table 1.** The four types of NN simulated (input & output parameters).

| | VERSION 1 | VERSION 2 | VERSION 3 | VERSION 4 |
|---|---|---|---|---|
| INPUT | $T$<br>$t$<br>$S_0$<br>$S$<br>$X$ | $T$<br>$t$<br>$S_0$<br>$S$<br>$X$<br>$pH$<br>$CO_2$ released | $T$<br>$t$<br>$S_0$<br>$X$ | $T$<br>$t$<br>$S_0$<br>$X$<br>$pH$<br>$CO_2$ released |
| OUTPUT | $P$ | $P$ | $P$<br>$S$ | $P$<br>$S$ |

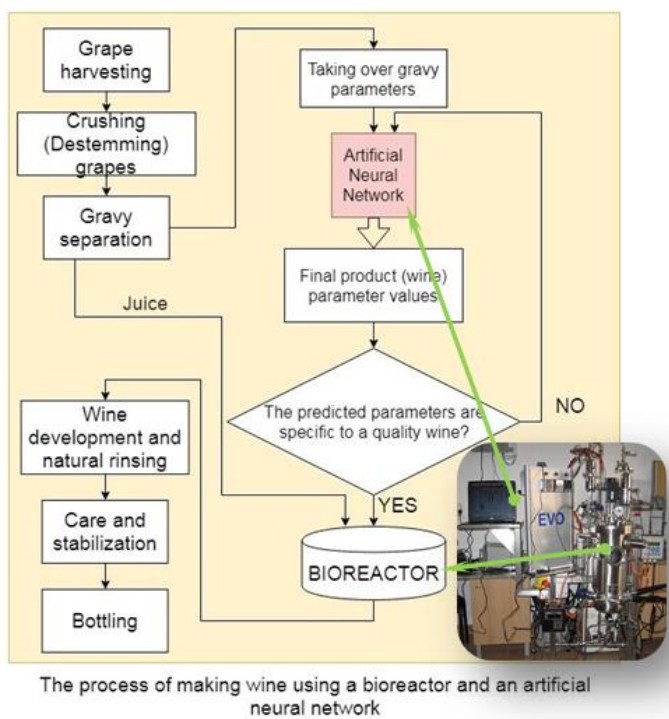

The process of making wine using a bioreactor and an artificial neural network

**Figure 4.** The scheme of using the NN in the technological process of white winemaking.

### 3.1. User Guide and System Requirements

The software application used to predict the variables of the alcoholic fermentation process was realised in the MATLAB environment. The graphical user interface (GUI) illustrated in Figure 5 was designed in a user–friendly way, simplifying access to non-specialists in computer science.

The graphical elements of the software application interface include several features. The radio buttons are used to choose which type of NN will be used for training: a feedback or feed-forward network. Through this software application, the user can accomplish the following:

- Choose an Excel (*.xlsx) file in which the training data of the network are structured
- Choose the type of network to predict the variables
- Set up the iteration number that the network will execute in the training process
- Set up the neuron number for the hidden layer of the network
- Set up the hidden layer number of the network
- Comparatively visualize graphics of the desired output of the network and the real output (see Figure 6)
- Predict the desired variable functions of several variables defined as the input network.

From a hardware point of view, the application needs to run properly on systems with quad-core processors, 8 GB RAM, and 4–6 GB of HDD space.

We created an archive with the sources of the applications, which can be accessed at the following web address http://webspace.ulbsibiu.ro/adrian.florea/html/simulatoare/AlcoholFermentationApplication.zip and downloaded to local computers by any interested parties. Further details about the software's use are provided by the README.txt file in the archive. The software application was developed using the full MATLAB campus license (MATLAB, Simulink and learning resources) provided for academic use, free of charge by "Lucian Blaga" University of Sibiu with the full support of "Hasso Plattner" Foundation Germany. For this reason, those who use our source must do so only for educational purposes. Besides being free and easy to use, our tool provides the following advantages: flexibility, extensibility, interactivity, and performance.

**Figure 5.** The graphical user interface of the software application.

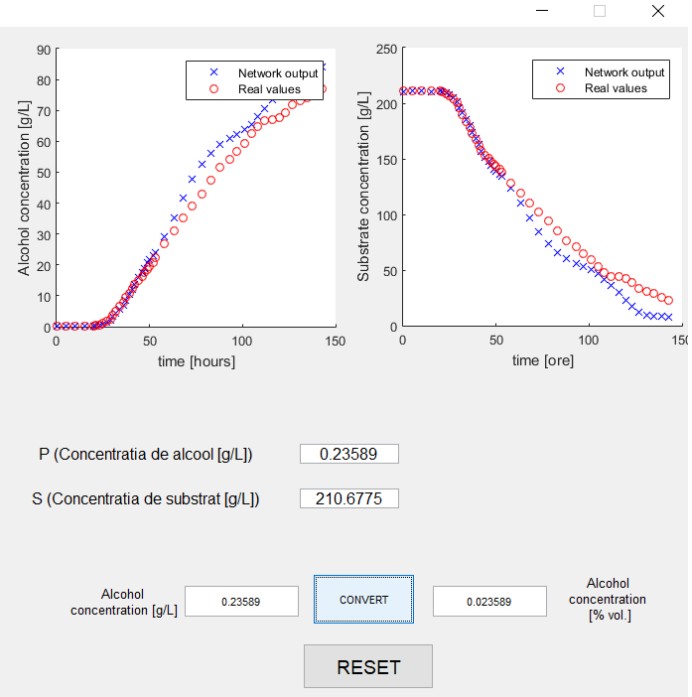

**Figure 6.** A comparative graphics between the desired output of the network and the real one.

### 3.1.1. The Training Phase

The NN training phase presumes the parsing of several experimental data (70% of the data), the generation of several outputs based on the input data, and adjustment of the weight values of the network as a function of the resulting errors among these outputs and the desired outputs from the experimental data set.

At the end of the training phase, the graphical interface will display the errors obtained from training data. These errors are calculated with the following equation:

$$E = \frac{\sum_{i=1}^{M}(NetworkOutput - ScopeValue)^2}{M} \tag{1}$$

where *M* is the pairs number of the input–output used in training.

### 3.1.2. The Testing Phase

After NN training phase comes the testing stage. This phase involves the use of several experimental data (the other 30% of the experimental data) and generating several outputs using a NN trained with those data. Then, the outputs are compared with the desired outputs from the experimental data. Based on the differences between these outputs, the errors of the network's predictions of these values are calculated.

## 4. Results and Discussion

We simulated several situations with a range of the maximum number of iterations and neurons from the hidden layer/layers to determine the best configuration of the NN. Moreover, we used the first version of the neural network from Table 1 and then tested the other versions. Each version of the training data has two variants. The first variant contains the original training data without any change from the experimental data, and the second version consists of the training data normalised with the following equation:

$$value_{normalised}(i) = \frac{value(i) - minimum}{maximum - minimum} \tag{2}$$

where $value_{normalised}(i)$ is the normalised value corresponding to the element with the *i* index from the original sequence, $value(i)$ is the value of the element with the *i* index from the original sequence, and $minimum/maximum$ is the minimum/maximum value from the original sequence.

In the realised simulations, using the fourth variant of the NN from Table 1, we varied the following parameters: the number of neurons from the hidden layer, the number of the iterations used for network training, the type of the NN, and the number of hidden layers. For each configuration, the application was run 10 times to obtain the medium value in each case because after the each run, the error values were different (the initial values of the NN weights were randomly generated). The next simulations were performed using a single hidden layer neural network and the Levenberg–Marquardt learning algorithm.

Figure 7 presents the results of the simulations obtained with the file that contained the original training data (not normalised).

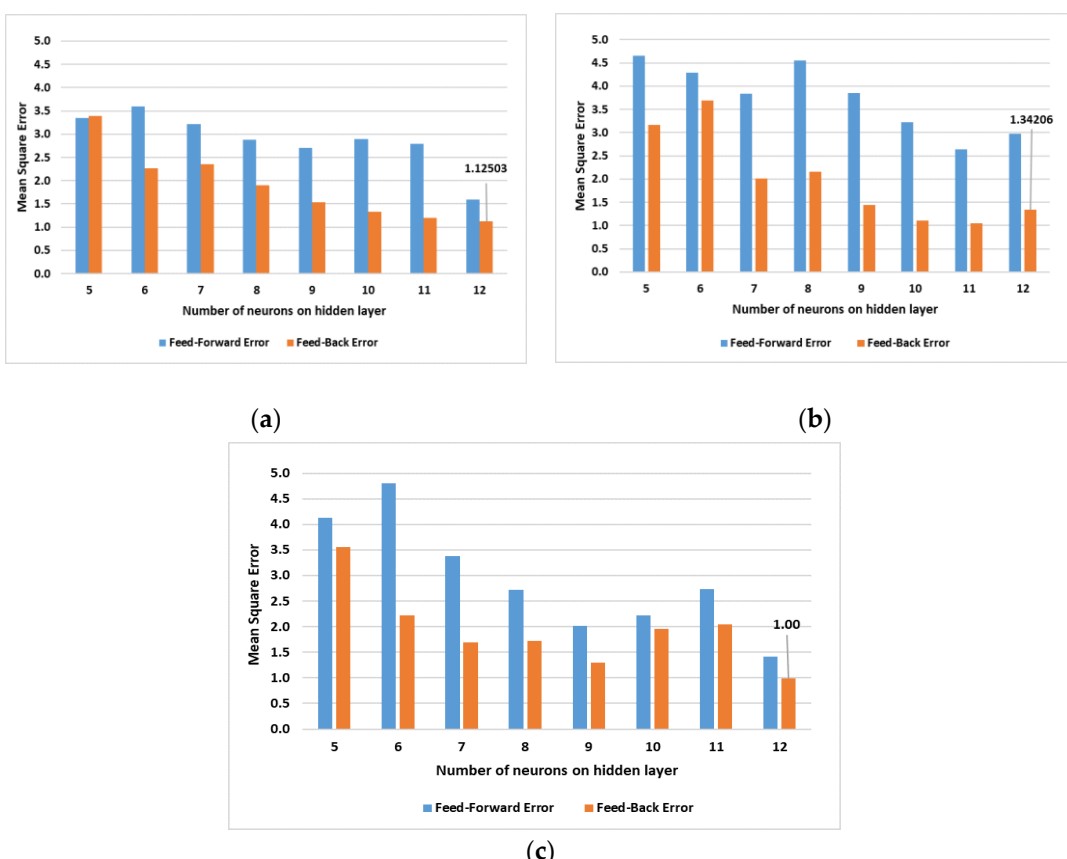

**Figure 7.** The obtained results by varying the number of neurons on hidden layer for (**a**) 500; (**b**) 1000 and (**c**) 1500 iterations.

As can be observed in the above illustrations, the simulation that used 1500 iterations for the feedback network training with a hidden layer and 12 neurons had the lowest error. Thus, for the following predictions, we used a network trained in this way.

For the simulations with the file that contained the original training data and normalised data by varying number of neurons on hidden layer for 1500 iterations the results are presented in Figure 8.

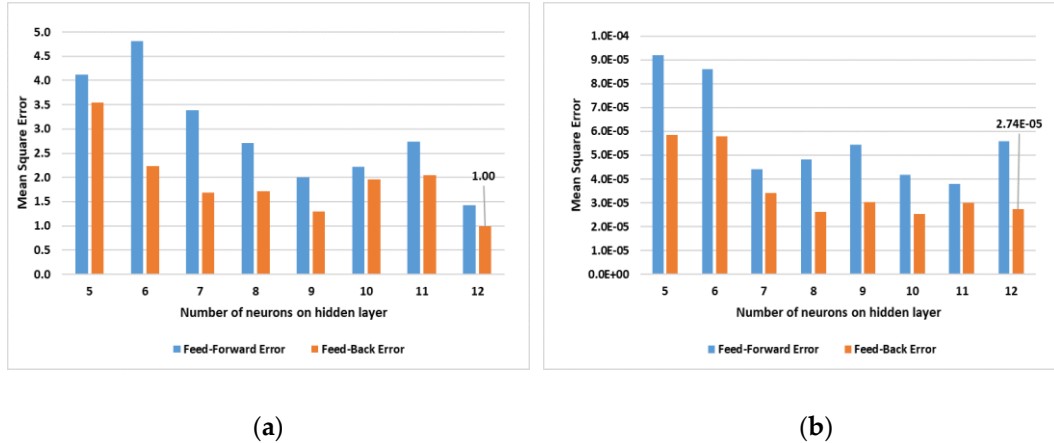

(**a**)                  (**b**)

**Figure 8.** The obtained results by varying number of neurons on hidden layer for 1500 iterations: (**a**) original data; (**b**) normalised data.

Furthermore, we simulated feedback NNs by varying the number of hidden layers and the learning algorithms. Based on these simulations, it was concluded that using one hidden layer in the NN is the best solution because the simulation time increased significantly with greater numbers of hidden layers.

Table 2 presents the simulation results using the Levenberg-Marquardt and Backpropagation algorithms, respectively. The best results with the lowest errors were generated with the Levenberg-Marquardt algorithm.

**Table 2.** Simulation results obtained with the original data using the Levenberg-Marquardt and the Backpropagation algorithms respectively.

| No. of Iterations | No. of Hidden Layers | No. of Neurons on Hidden Layers | Average Error | |
|---|---|---|---|---|
| | | | Levenberg-Marquardt Algorithm | Backpropagation Algorithm |
| 1500 | 1 | 5 | 3.5498 | 5.6236 |
| 1500 | 1 | 6 | 2.2282 | 5.4599 |
| 1500 | 1 | 7 | 1.6872 | 4.5100 |
| 1500 | 1 | 8 | 1.7170 | 3.9243 |
| 1500 | 1 | 9 | 1.3018 | 4.6077 |
| 1500 | 1 | 10 | 1.9558 | 5.9983 |
| 1500 | 1 | 11 | 2.0424 | 3.8234 |
| 1500 | 1 | 12 | 0.9954 | 4.0962 |

By analysing the above graphics and tables, it can be concluded that the normalised data generate the lowest Mean Square Error (MSE). However, even though the error is lower with normalized data, this is not significant because from a technological point of view, normalization does not change anything. Thus, the next simulations used the original data (without normalization).

Based on all the simulations presented, the optimal configuration for the NN is as follows:

- The network type: feedback
- The number of iterations: 1500
- The number of hidden layers: 1
- The number of neurons from the hidden layer: 12
- The learning algorithm: Levenberg-Marquardt.

In the following section, we present graphics with results obtained using all four variants of the neural networks specified in Table 1. These graphics illustrate a comparison between the NN output used for the tests and the experimental data.

For the first three versions of the training and testing data in Table 1, all the used values were expressed at the same temperature. For the fourth version, the data set used for training and testing contained values expressed as multiple temperatures. In the next figure (Figure 11), each temperature is graphically represented. For the fourth version from Table 1, in Figure 11 (a1, b1, c1, and d1), a graphic illustrating the error between the NN output and the experimental data is provided for each temperature.

By comparing the first two variants of NNs specified in Table 1 from perspective of the alcohol concentration prediction in accordance with the real fermentation process (see Figure 9), the best NN configuration was found to be the second version because its latent phase is shorter, which is desirable. Furthermore, the alcohol concentration evolution depends on the yeast type, the medium characteristics, the substrate concentration, and the temperature. In variant two, we introduced two additional variables as inputs in the training process: the pH and $CO_2$ released concentration. The exponential growth phase in the first variant begins after 131 h in comparison with the second variant, which begins after 30 h.

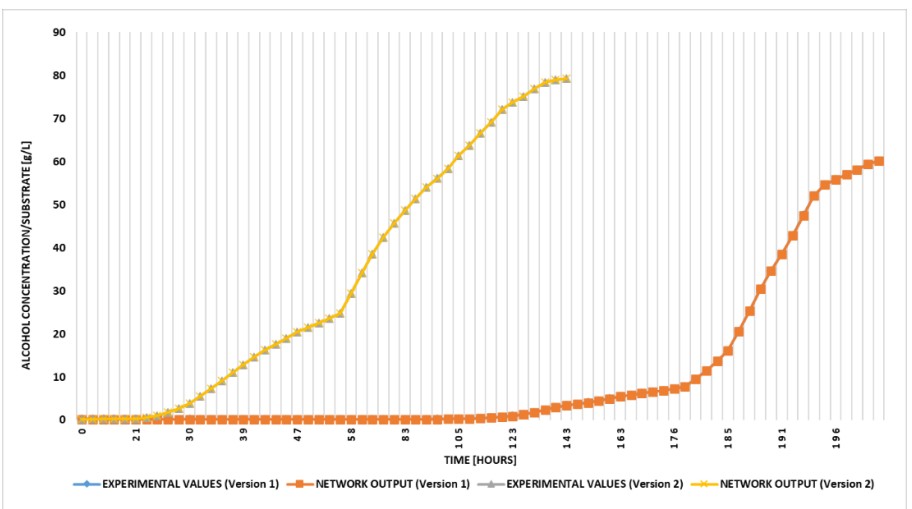

**Figure 9.** Generated results by the NN vs. experimental values using the first and second variants input-output configurations of NN.

Figures 10 and 11 also illustrate the prediction of two parameters: the substrate concentration and the alcohol concentration. Based on the substrate consumption, the alcohol is produced. Moreover, in Figure 11, the NN used the pH values and released $CO_2$ concentration as additional inputs; in this case, the training and testing data also contain values from multiple fermentation processes (different temperatures).

The illustrations in Figure 10 are similar to those in Figure 11. Based on all these combinations (Figure 11), the best results are obtained at a 20 °C fermentation process temperature, at which the NN output is almost identical with the experimental data (Figure 11a2), and the errors also have the smallest values (Figure 11a1).

In all three Figure 9, Figure 10, Figure 11, the results show that the NN variants that used the pH values and released $CO_2$ concentration from the training data generated better results regardless of the output number.

Since a comparison of our results with other papers would be inadequate due to the lack of an international framework with standardized benchmarks, we chose to compare the data obtained from the simulations with an analytical model developed in a previous work by one of the authors to validate the artificial intelligence algorithms used.

In addition, because our tool is very flexible, we made some comparisons with learning methods (NN with feed forward versus feedback or varying the number of layers and the number of nodes in each layer).

The results also show a good fit with the nonlinear mathematical model of the batch wine alcoholic fermentation process previously developed by Sipos et al. [25]. The evolution of the concentrations of substrate and alcohol predicted by the model are similar to the observed data (Figure 12). By graphically comparing the experimental results in Figures 9–11 alongside the evolution resulting from the analytical model in Figure 12, we validated the power of the neural networks, which are entirely data-based and require no previous knowledge of the events that govern the process and are available for learning, analysis, association, and adaptation. For completeness, the equations of the model developed by Sipos et al. considering the latent and exponential growth phases are as follows:

- biomass:

$$dX/dt = \mu_{max}\cdot(S/(K_S + S))\cdot \exp(-K_P\cdot P)\cdot X \qquad (3)$$

- alcohol:

$$dP/dt = q_{pmax}\cdot(S/(K_{SP} + S))\cdot \exp(-K_{pp}\cdot P)\cdot X \qquad (4)$$

- substrate:

$$dS/dt = -(1/Y_{XS}\cdot dX/dt) - (1/Y_{PS}\cdot dP/dt) \qquad (5)$$

- $CO_2$ released:

$$(dC_{CO_2})/dt = g\cdot C_{CO_2}\cdot k\cdot S/(K_{SP} + S)\cdot \ln(k\cdot S/(K_{SP} + S)\cdot t) \qquad (6)$$

where $\mu_{max}$ represents the maximum specific growth rate [1/h], $K_S$ is the substrate limitation constant [g/L], $K_P$ is the alcohol limitation constant [g/L], $q_{pmax}$ is the maximum specific alcohol production rate [g/(g·cells·h)], $K_{SP}$ is the constant in the substrate term for ethanol production [g/L], $K_{pp}$ is the constant of fermentation inhibition by ethanol [g/L], $Y_{XS}$ is the ratio of cells produced per the amount of glucose consumed for growth [g/g], $Y_{PS}$ is the ratio of ethanol produced per the amount of glucose consumed for fermentation [g/g], $g$ is the pseudo-constant of $CO_2$, and $k$ is the kinetic constant [1/h].

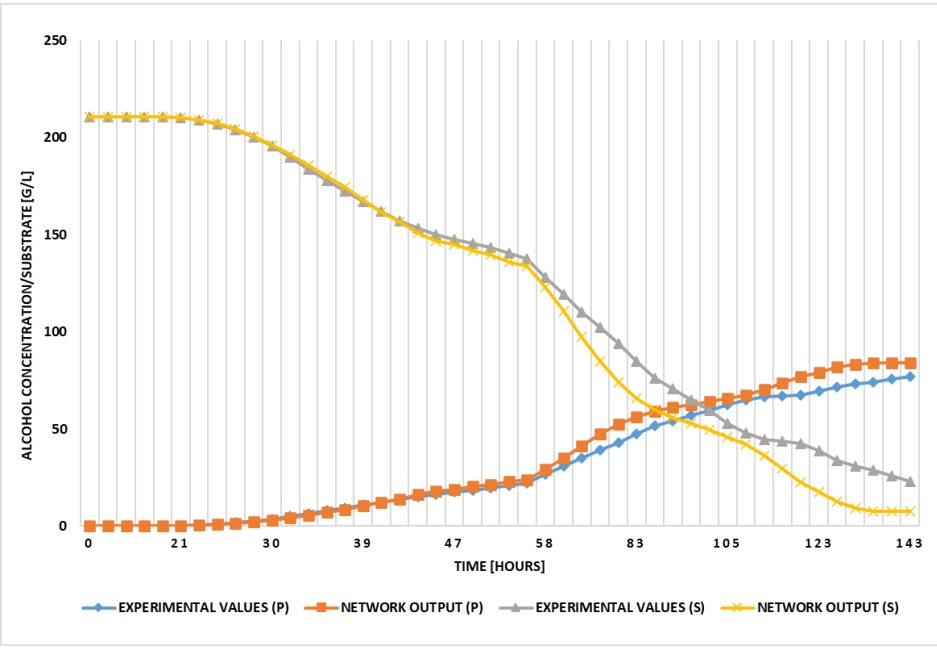

**Figure 10.** Generated results by the NN vs. experimental values using the third variant input-output configuration of NN.

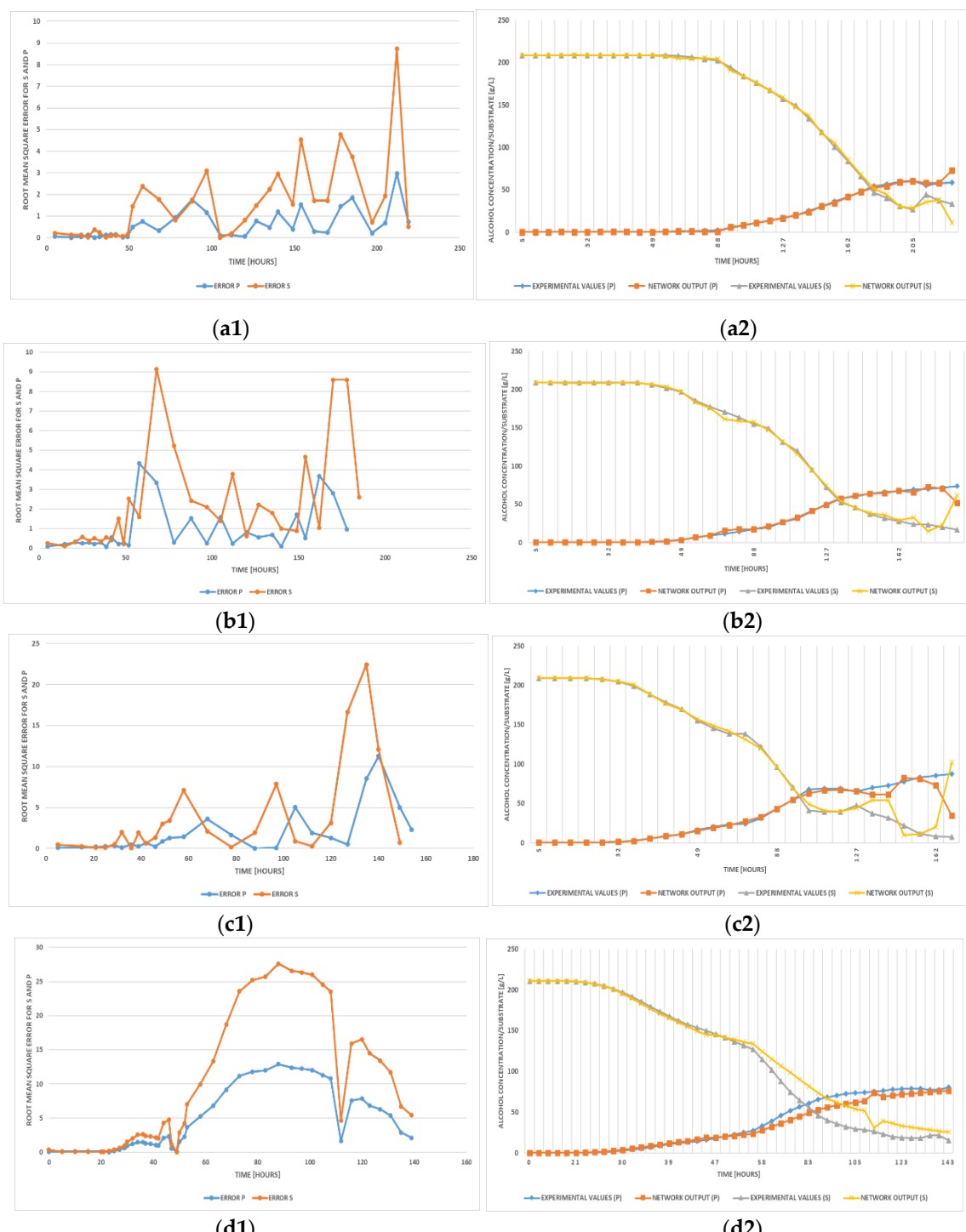

**Figure 11.** MSE (**a1**–**d1**) and generated results by the NN vs. experimental values (**a2**–**d2**) using the fourth variant input-output configuration of NN at 20, 24, 26 and 28 °C.

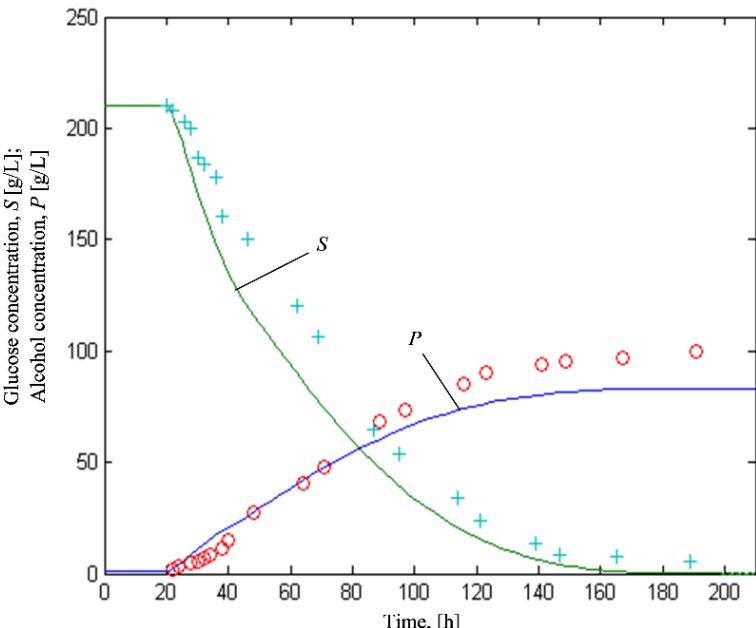

**Figure 12.** Evolution of glucose and alcohol concentrations; a comparison between experimental values ((**o**)—glucose and (**+**)—alcohol) and simulation results (continuous lines).

*Implications for Practice*

In the following section, we describe a few advantages of our work. Based on the graphically illustrated results, the proposed solution is useful at least at the educational level when (masters) students are presented with the advantages of software applications/sensors in the fermentation process. The advantages include the following:

- faster identification of the moment of transition from the latent to the exponential phase
- the use of additional process parameters such as pH and $CO_2$ as inputs to the neural network determining an improvement in the quality of the predictions
- the role of temperature in accelerating the fermentation
- the need for a rich set of data and test sets that are as standardized as possible
- understanding the differences between the learning mechanisms used
- the simple extensibility of the application by using genetic algorithms as a training method or for the automatic design of the optimal structure of a neural network.

Moreover, for the companies that produce wine, with the help of this application, the evolution of the quantitative characteristics of the alcoholic fermentation process for white wines can be predicted. The purpose of predicting these process parameters is to obtain information on state variables to automatically drive the process, especially when these parameters cannot be directly measured.

## 5. Conclusions

This paper explored the development of an application by engineers in the food industry for an alcoholic fermentation process that will reduce the laboratory measurement time of the variables that characterise this process and provide a substitute for the sensors that measure these variables, thereby increasing the lifecycle of sensors from the bioreactor. By using this application, the specialists in this domain can simulate many situations that emerge during the fermentation process and thus optimise the technological process without using an industrial pilot.

Indirect information about the state variables in the alcoholic fermentation process was determined after manual design space exploration of the structure, type, and inputs

of a multilayer neural network. The conclusion was that the recurrent (feedback) NN generated the best results. The optimal configuration that creates the minimum error for this NN is 1500 iterations, 7 neurons on the input layer, one hidden layer, and 12 neurons on the hidden layer. The second and fourth variants provided in Table 1, which outlines the pH and $CO_2$ concentrations released as inputs in the neural network, produced the best results.

Further directions that use or improve this application could involve the following:

- A large dataset could be used for experiments or for other fermentation processes in winemaking, while other machine learning techniques like Support Vector Machines or Gauss Process Regression [14] could be explored.
- Due to the properties of the raw material's variability together with the process's complexity and the nonlinearity and the non-stationary of the variables that characterise the fermentation process, designing an appropriate control system will be adequate using a system based on knowledge. The real-time information that led to automation control can be derived from the process's physical transducers, as well as from the state observers and the developed NN application.

We intend to improve the neural network prediction accuracy by including prior knowledge in the NN and by employing training mechanisms. One such direction involves the implementation of Genetic Algorithms (GAs) to improve the performance of the neural network. Gas can be used in two contexts: the first aims to find a vector of optimal weights and biases for the initial configuration of the network instead of random weights; the second context aims at determining the optimal structure of the neural network through an automatic (not manual) design space exploration process—namely, the optimal number of levels in the hidden layer, the number of nodes in each hidden layer, the appropriate activation function, and the network type (feed forward/feedback).

**Author Contributions:** All authors contributed equally to the research presented in this paper and to the preparation of the final manuscript. All authors have read and agreed to the published version of the manuscript.

**Funding:** This research received no external funding.

**Data Availability Statement:** The data presented in this study are available on request from the corresponding author. The data are not publicly available because they were obtained in a Ph.D. thesis.

**Conflicts of Interest:** The authors declare no conflict of interest.

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
