# Peer review of "Using Neural Networks to Obtain Indirect Information about the State Variables in an Alcoholic Fermentation Process"

_processes, doi:10.3390/pr9010074_

Round 1
Reviewer 1 Report
Sipos et al. presented an artificial neural network modeling approach to describe the alcoholic fermentation process, which is important in wine industry. Specifically, the authors developed a black box model based on the process data to predict the key parameters, which can be useful for the process control. Although the results seem promising and potentially interesting to readers, I have a few comments to be addressed by the authors
- It seems that there are already a number of publications on developing ANN for modeling alcoholic fermentation process. For example, Mjialli and Al-Asheh ( https://doi.org/10.1002/ceat.200500166); (https://doi.org/10.1016/j.engappai.2008.06.001); Mantovanelli et al. (https://doi.org/10.1016/S1570-7946(06)80113-6). What are the novelty of this work compared to others, and which unanswered challenges the authors are able to address?
- The resolutions of the figures need to be improved. As they are, they are quite blurry.
- The authors showed the GUI that streamlined the process. It will be extremely helpful and useful if the authors can provide a link where the users can find and download this software.
- In the discussion, the authors presented an ODE model for the alcoholic fermentation process, which the authors developed in their previous study. I am not sure why the authors presented this at the end of this work, which seems irrelevant to the scope of this manuscript.
- Related to the first comment, if the authors already have a fairly good mechanistic model, it seems that a more appropriate choice is to develop a hybrid mechanistic-neural network model. For example, Thompson and Kramer, https://doi.org/10.1002/aic.690400806; Lee et al., https://doi.org/10.1371/journal.pcbi.1008472 ; Bangi and Kwon, https://doi.org/10.1016/j.compchemeng.2019.106696 ; Teixeira et al., https://doi.org/10.1016/j.jbiotec.2007.08.020). Specifically, one can retains the mechanistic model and combines with the neural network model to increase the accuracy.
There are some grammatical mistakes need to be addressed. For example
- In the abstract, "that helps" should be changed to "to help" in the line 14.
- In the abstract, "s" should be removed after "increase" in the line line 15
- The sentence from 175 to 181 is a very long run-on sentences, which needs to be corrected.
- In the line 202, it is not clear what the sentence "... white graph must and white graph must enrich with B1..." tries to convey.
- In the line 290, there is no noun for the sentence starting with" has been used the.."
- Add a comma after "experimental data" on line 293
The last comment I have is about the references. Some references have doi while other do not. Please be consistent.
Author Response
The modified and added texts are in red.

Reviewer 2 Report
Regarding this article, I have a few remarks:
In general, the purpose of the article is formulated at the end of the introduction. The goal is missing in this article. An aim is formulated in the conclusions section. In addition, the goal should correspond closely with the title of the article!
The results are not clearly presented. Figures 3, 4, 5,6, 7 are not clear and are difficult to read.
The conclusions are more like a summary of this article, and should reflect the most important scientific conclusions of the authors.
Here is an example of a specific and accurate conclusion taken from a paragraph in this article:
This paper explores the development of an application by the engineers from food industry, on an alcoholic fermentation process, which will reduce the laboratory measurements time of the variables that characterise this process and also can substitute the sensors that measure these variables, increasing the lifecycle of sensors from bioreactor.
Recommendations for the practice:
By using this application, the specialists in this domain can simulate many situations that appear during a fermentation process and can optimise the technological process without using an industrial pilot.
Further directions of using or improving this application can be:
• large dataset can be taken for experiments, for other fermentation processes in winemaking and other NN learning techniques may be explored;
• due to the properties of the row material variability together with the process complexity, nonlinearity and non-stationary of the variables that characterise the fermentation process, designing an appropriate control system will be adequate using the system based on knowledge. The real time information that led to the automation control can be derived from the process’s physical transducers as well as from the state observers and the NN application developed.
Author Response

(The authors gave the same response as above.)

Reviewer 3 Report
Manuscript ID processes-1056302, title: Using neural networks to obtain indirect information about the state variables in an alcoholic fermentation process. In this work, a controlling solution of the alcoholic fermentation process was developed using neural networks (NNs) in order to obtain indirect information about the state variables in the process.
The works presented in the manuscript have an application and practical character and correspond to the profile of the Prcesses journal. The manuscript contains valuable information and may provide guidance for operators of alcohol fermentation systems. However, it must be corrected and completed before being published.
In the abstract, the Authors should present the influence of the applied optimal configuration of NN with the obtained technological effects of the fermetation process. Stating which option is the best is definitely not enough. It needs to be completed.
Graphical abstract will be useful
Authors must change the manuscript to scientific works. As it stands, this is an operational report rather than well thought-out and planned scientific research.
The very long introduction provides only basic, commonly known information about NN and the application of this method in monitoring and control of technological processes. Lack of documenting the background, inspiration and the need for research and exploitation works carried out in the scope presented in the manuscript.
Chapter 2.1.1. The NNs types is unnecessary in my opinion. It is not a review. Likewise 2.1.2.
The scheme of organization of experiments and research works will be helpful in understanding the methodology. Division into stages, series, variances, options would be advisable.
Have the Authors used any methods of designing research works, e.g. Placket-Burman, surface response methodology, etc. On what basis were the analyzed research variants selected.
There is no discussion of the obtained results, no comparison with the results of other researchers. A profound comparative analysis showing the advantage of the applied solution is necessary.
On what basis was the best variant selected. There is no clear relationship with the efficiency of the alcoholic fermentation process in both the discussion of the results and the conclusions. Needs to be completed.
The article is carelessly written. The Authors must improve the English language and the presentation of the obtained research results. Line 195: The alcoholic fermentation process is a process….
Figure 3. It is completely illegible and needs to be corrected. Only the third part of the drawing should be left (The process of making wine using s bioractor and artificial NN)
Figure 4, figure 5 and figure 6, 7 are of very poor quality and need to be corrected.
Figure 8 is of very poor quality. Delete the data shown in the chart and add to the signature. The legend does not match the data in the chart no experimental values ​​(version 1) and experimental values ​​(version 2)
Figure 9 is of very poor quality. Delete the data shown in the chart and add to the signature.
Figure 10. Apply the notes above.
Figure 11. Something went wrong with the y axis description. Need to be corrected. The lack of a legend makes it difficult to interpret the results.
In my opinion, the manuscript in its current form cannot be published. It requires a deep correction and many changes and additions.
Author Response

(The authors gave the same response as above.)

Round 2
Reviewer 1 Report
The authors addressed my comments appropriately, so I recommend its acceptance for the publication.
Reviewer 3 Report
Thanks to the Authors for their responses to my comments and remarks. In my opinion, a revised version of the mnuscript can be published.